# Solvent-mediated assembly of atom-precise gold–silver nanoclusters to semiconducting one-dimensional materials

Peng Yuan[1,3], Ruihua Zhang[1,3], Elli Selenius [2], Pengpeng Ruan[1], Yangrong Yao[1], Yang Zhou[1], Sami Malola[2], Hannu Häkkinen [2✉], Boon K. Teo[1], Yang Cao[1✉] & Nanfeng Zheng [1✉]

Bottom-up design of functional device components based on nanometer-sized building blocks relies on accurate control of their self-assembly behavior. Atom-precise metal nanoclusters are well-characterizable building blocks for designing tunable nanomaterials, but it has been challenging to achieve directed assembly to macroscopic functional cluster-based materials with highly anisotropic properties. Here, we discover a solvent-mediated assembly of 34-atom intermetallic gold–silver clusters protected by 20 1-ethynyladamantanes into 1D polymers with Ag–Au–Ag bonds between neighboring clusters as shown directly by the atomic structure from single-crystal X-ray diffraction analysis. Density functional theory calculations predict that the single crystals of cluster polymers have a band gap of about 1.3 eV. Field-effect transistors fabricated with single crystals of cluster polymers feature highly anisotropic *p*-type semiconductor properties with ≈1800-fold conductivity in the direction of the polymer as compared to cross directions, hole mobility of ≈0.02 cm$^2$ V$^{-1}$ s$^{-1}$, and an ON/OFF ratio up to ≈4000. This performance holds promise for further design of functional cluster-based materials with highly anisotropic semiconducting properties.

[1] State Key Laboratory for Physical Chemistry of Solid Surfaces, Collaborative Innovation Center of Chemistry for Energy Materials, National & Local Joint Engineering Research Center of Preparation Technology of Nanomaterials, College of Chemistry and Chemical Engineering, Xiamen University, 361005 Xiamen, China. [2] Departments of Physics and Chemistry, Nanoscience Center, University of Jyväskylä, FI-40014 Jyväskylä, Finland. [3] These authors contributed equally: Peng Yuan, Ruihua Zhang. ✉email: hannu.j.hakkinen@jyu.fi; yangcao@xmu.edu.cn; nfzheng@xmu.edu.cn

The remarkable progress in the synthesis, structural discovery, functionalization and theoretical understanding of ligand-stabilized, atom-precise metal nanoclusters has opened fascinating opportunities to use these well-defined nanometer-size building blocks for designing nanomaterials with tunable properties[1–9]. The bottom-up fabrication of nanomaterials relies on spontaneous or directed self-assembly. Self-assembly based on supramolecular weak interactions or designed linkers was introduced for bottom-up fabrication of nanomaterials, consisting of colloidal metal nanoparticles, since mid-1990's. Early successful realizations included gold nanoparticle assemblies where individual particles were linked by dithiols[10], DNA[11,12], or using viral proteins as scaffolds[13,14]. Gold nanoclusters stabilized by thiolates[15] have for a long time served as a prototype, and the first successful structural determinations of their atomic structures led to breakthroughs in understanding the structure of the gold–ligand interface, the interactions in the ligand layer, the organic surface, and the principles that govern their electronic structure and optical response[16–21], This has inspired efforts towards nanoscale and macroscale assemblies of gold[22–26], silver[27,28], and copper clusters[29–32] that might have a variety of applications in the fields of (bio)chemical sensing[33–35], and optoelectronics[29–32,36–38].

Up to date, cluster self-assembly has been demonstrated mostly for three-dimensional (metal-organic framework-like or spherical capsids) and 2D (film) systems, while successful realizations of anisotropic one-dimensional (1D) cluster materials have been scarce until very recently[26,38]. Here, we demonstrate a solvent-mediated assembly of 1-ethynyladamantane (A-Adm) protected, intermetallic 34-atom Au–Ag clusters (hereafter labeled as $(AuAg)_{34}$ and A-Adm noted as ligand L) into cluster polymers that feature direct metal–metal bonds connecting the clusters into 1D chains. This material makes macroscopic single crystals with anisotropic semiconducting properties. Density functional theory (DFT) calculations of the crystal predict semiconductor-type electronic band structure with a band gap of about 1.3 eV. This material is amenable for testing as a device component for a field-effect transistor (FET) showing p-type behavior, highly anisotropic electrical conductivity with about 1800-fold difference along/across the polymer chain, hole mobility of ≈0.02 cm$^2$ V$^{-1}$ s$^{-1}$, and an ON/OFF current ratio up to 4000 of the FET device.

## Results

### Synthesis of the $(AuAg)_{34}$ clusters and observation of polymerization.
The syntheses of $(AuAg)_{34}$ and $(AuAg)_{34n}$ nanoclusters in different solvents are summarized in Fig. 1. In a typical synthesis, AuSMe$_2$Cl was first dissolved in a mixture solvent of chloroform and methanol. A-Adm was then added to the solution and stirred for 20 min. Subsequently, Ag acetate and tert-butylaminoborane were added to the mixture under vigorous stirring. The reaction was aged for 10 h at room temperature. During the period, the color of the reaction mixture gradually changed from yellow to dark red. The solution was then centrifuged for 2 min at 10,000 r min$^{-1}$ to give a dark red solution. After being stored at 25 °C, black block crystals were formed in ca. 7 days. When dichloromethane was used in otherwise similar synthesis, subsequent crystallization yielded single crystals containing the cluster polymers. Furthermore, we observed that when the crystals of the monomeric $(AuAg)_{34}$ were re-dissolved in a mixture of methanol and CH$_2$Cl$_2$, crystals containing the cluster polymers were obtained.

It is intriguing to observe such dramatic effect in our synthesis caused by an apparently minute change in the synthesis conditions by two closely related solvents. On the other hand, it is well documented that solvent can play a pivotal role in the

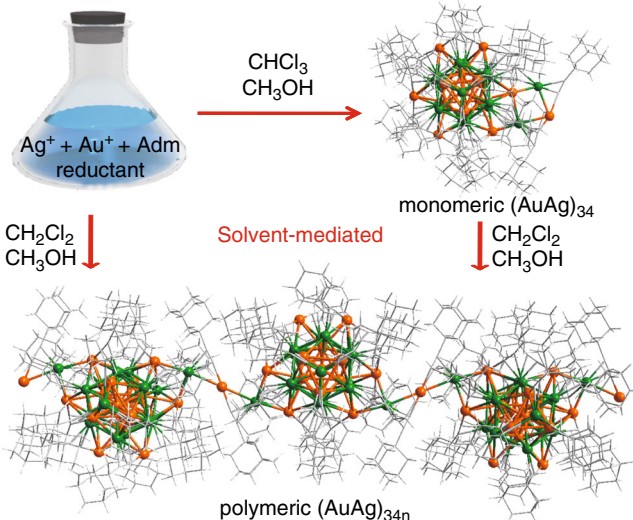

**Fig. 1 Schematics of solvent-mediated self-assembly of the cluster polymers.** Depending of the solvent, either monomeric clusters or cluster polymers are formed. Colors: golden and green, Au/Ag; gray, C. All hydrogen atoms are omitted for clarity.

synthesis of metal nanoclusters[39,40]. Changing the chemical environment can drive structural transformation of Au/Ag nanoclusters affecting their physico-chemical properties[41–46].

### Atomic structure of the building block $(AuAg)_{34}$ and the cluster polymer.
The total structures of $(AuAg)_{34}$ and the cluster polymer $(AuAg)_{34n}$ were determined by single-crystal X-ray diffraction. Both of $(AuAg)_{34}$ and $(AuAg)_{34n}$ crystallize into C2/c space group (crystal data shown in Supplementary Table 1). Similar to the cases of thiolate-protected $(AuM)_{38}$ (M = Ag or Cu) nanoclusters[47–50], positional disorders of Au and Ag atoms also occur in the kernels of $(AuAg)_{34}$ and $(AuAg)_{34n}$. As shown in Supplementary Tables 2 and 3, based upon least-squares refinement of the X-ray data, the compositions of $(AuAg)_{34}$ and $(AuAg)_{34n}$ are Au$_{21.3}$Ag$_{12.7}$L$_{20}$ and [Au$_{21.4}$Ag$_{12.6}$L$_{20}$]$_n$, respectively. In addition, we have used inductively coupled plasma mass spectrometric (ICP-MS) and energy-dispersive X-ray spectroscopic (EDS) to get insight into their metallic distributions. The X-ray results are confirmed by independent compositional assignments made using of ICP-MS (1:1.69 and 1:1.73 molar ratio of Ag/Au for total 34 metallic atoms in $(AuAg)_{34}$ and $(AuAg)_{34n}$ respectively) and EDS (1:1.67 and 1:1.74 molar ratio of Ag/Au for total 34 metallic atoms in $(AuAg)_{34}$ and $(AuAg)_{34n}$, respectively) (Supplementary Figs. 1 and 2; and Supplementary Table 4).

The metal frameworks of $(AuAg)_{34}$ and $(AuAg)_{34n}$ are described in Fig. 2 and Supplementary Figs. 3, 4, 5a, 5b. They are very similar, differing only by the decoration of the outermost three atoms, namely, AuAg$_2$ (Fig. 3). The anatomy of the metal structure of $(AuAg)_{34n}$ is shown in Fig. 2. It is interesting to note that $(AuAg)_{34}$ and $(AuAg)_{34n}$ share the same $(AuAg)_{31}$L$_{18}$ unit (Fig. 2 and Supplementary Figs. 1 and 5). Specifically, $(AuAg)_{31}$ can be described as bi-centered-icosahedra sharing a fusion M$_3$ face (13 × 2 – 3 = 23), capped with a hexagonal ring of exterior shell Au$_6$ in the "equatorial" plane of the shared fusion M$_3$ face and two apical Ag atoms (Fig. 2). The metal site Au/Ag disorders, as obtained by least-squares refinement of the occupancies, are indicated in Supplementary Figs. 1 and 5b and Supplementary Tables 2 and 3. Finally, the exterior metal shell comprises an Ag–Au–Ag unit (Fig. 3 and Supplementary Figs. 4b, c and 5b) with one Ag atom bridging one of the nonbonding edges of the "equatorial" hexagonal ring of exterior shell Au$_6$ (Fig. 3). The

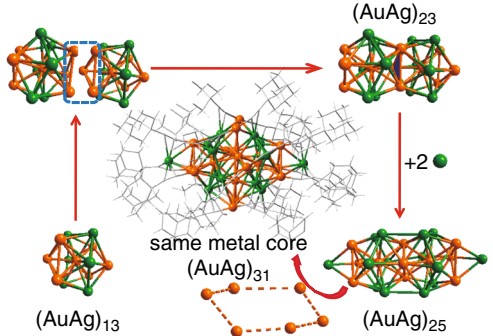

**Fig. 2 Anatomy of the core structure of (AuAg)₃₄ and (AuAg)₃₄ₙ.** Colors: golden and green, Au/Ag; gray, C. All hydrogen atoms are omitted for clarity.

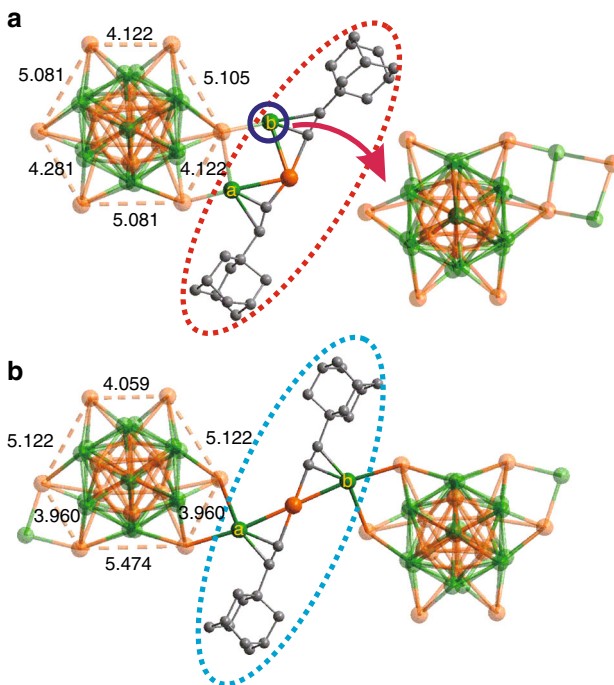

**Fig. 3 Different "Ag-L-Au-L-Ag" units of (AuAg)₃₄ and (AuAg)₃₄ₙ.** The Au-Au distances in the distorted Au₆ hexagon and Ag-Ag distance in the "Ag-Au-Ag" unit of **a (AuAg)₃₄** and **b (AuAg)₃₄ₙ**. Colors: golden and green, Au/Ag; gray, C. All hydrogen atoms are omitted for clarity.

Ag–Au–Ag unit is bent in the monomeric **(AuAg)₃₄** (Fig. 3a and Supplementary Figs. 4b and 5b) but linear in the polymeric **(AuAg)₃₄ₙ** (Fig. 3b and Supplementary Fig. 4c), which holds the key to the structural transformation from **(AuAg)₃₄** to **(AuAg)₃₄ₙ** (vide infra). The Ag–Au–Ag unit in the monomeric nanocluster is disorder because of the symmetry of the crystal structure (Supplementary Fig. 5b). For the three possibilities of the nearest adjacent nanoclusters in the real unit cell, the nearest Au/Ag–Au/Ag distances should be 5.348, 6.814, and 10.241 Å, respectively (Supplementary Fig. 5c–e). The Au atom in the Ag–Au–Ag linkage of the polymeric **(AuAg)₃₄ₙ** nanocluster is disordered with 5 possible positions arranged in a same plane (Supplementary Fig. 6). In comparison, because of their π-type bonding with Ag, the disordering of the associated ligands is not significant.

A detailed structural analysis of the bi-icosahedral **(AuAg)₂₃** cores in both **(AuAg)₃₄** and **(AuAg)₃₄ₙ** is shown in Supplementary Fig. 7. Metal–metal bond lengths between the central atom and the first-shell M (M = Au or Ag) atoms range from 2.71 to

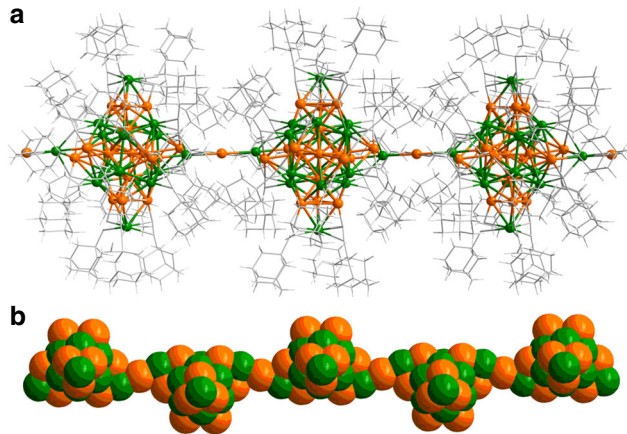

**Fig. 4 Atomic structure of the cluster polymer. a** Total structure viewed approximately orthogonal to the c-axis (parallel direction). **b** Space-filling view of the metal atoms (viewed from the top direction in **a**). Colors: golden and green, Au/Ag; gray, C. All hydrogen atoms are omitted for clarity.

2.83 Å, whereas the M···M (M = Au or Ag) distances in the icosahedral shells (including M atoms in the shared triangular face) are in the range of 2.77–3.12 Å. Between the two M₆ (M = Au or Ag) corrugated planes, three nearest M···M (M = Au or Ag) pairs are significantly longer (average 3.28 Å) and are evenly distributed at the waist of the **(AuAg)₂₃** rod. Bond lengths between each exterior Ag atom and the Au₃ (100% Au occupancy) faces range from 2.95 to 3.02 Å (2.99 Å (av) for **(AuAg)₃₄ₙ** and 2.98 Å (av) for **(AuAg)₃₄**).

In the plane defined by the shared M₃ triangle, six exterior shell Au (100% Au occupancy) atoms are bonded, respectively, to the three M atoms of the fusion plane in a radial fashion, with an average M···M distance of 3.08 Å. These six exterior Au atoms form a distorted hexagon with alternating long and short nonbonding Au···Au distances (5.24 (av) and 4.00 (av) Å for **(AuAg)₃₄ₙ**; 5.09 (av) and 4.18 (av) Å for **(AuAg)₃₄**). The latter (short) distances play a key role in the **(AuAg)₃₄** to **(AuAg)₃₄ₙ** transformation, as we shall see next (Fig. 3).

There are also 18 ligands on the surface of **(AuAg)₃₁** metal framework. Every Au atom of the hexagonal arrangement Au₆ shell is linearly bound by two ligands (Supplementary Fig. 4b). Each "apical" Ag atom is coordinated to the C–C π-bonds of three ligands (Supplementary Fig. 4a).

The key difference between **(AuAg)₃₄** and **(AuAg)₃₄ₙ** lies in the connectivity of the cluster building blocks of (AuAg)₃₁L₁₈ via the "linker hinge" unit "Ag(a)–L–Au–L–Ag(b)" highlighted in Fig. 3. In the monomer **(AuAg)₃₄** (top), one end of the linker, Ag (a), is anchored on a short edge of the exterior (nonbonding) Au hexagon of a given cluster (left). The other end, Ag(b), instead, bends back to bind to an adjacent Au atom of the same cluster unit, giving rise to an Ag(a)–Au–Ag(b) angle of 84.50 degree. Upon transformation to the polymer form, the Ag(a)–Au–Ag(b) angle of the linker is straightened to 180.00 degree, with the Ag(b) atom being inserted into the short Au···Au edge of the exterior (nonbonding) Au hexagon of the nearest adjacent cluster unit (right). This chain reaction propagates to produce the polymer **(AuAg)₃₄ₙ**. As a result, each (AuAg)₃₁L₁₈ cluster unit in the polymer has two linear linker hinges attached to it as shown in Fig. 4). In the single crystal, the polymeric chains of **(AuAg)₃₄** clusters are stacked in the unit cell as shown in Fig. 5, indicative of highly anisotropic crystal properties as will be discussed later.

Contrasting the two structures, given the exact same chemical formulations and virtually identical (AuAg)₃₁L₁₈ building blocks, the driving force for the monomer to polymer transformation can

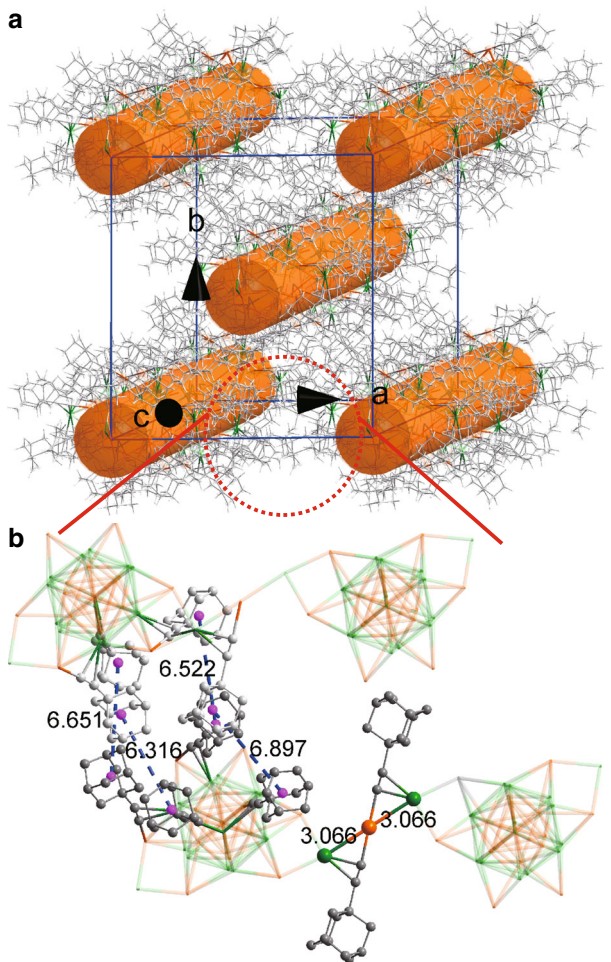

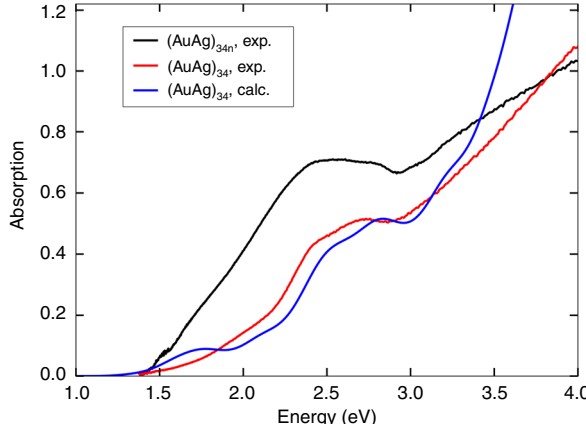

**Fig. 6 UV–Vis absorption spectra.** Absorption spectra of both monomeric and polymeric crystals. Computed spectrum for one of the isolated $(AuAg)_{34}L_{20}$ model clusters (see details of the models in the Methods) is shown as well. The intensity of the computed spectrum is scaled to the experimental peak at 2.7 eV of the monomeric cluster but no shifts have been applied in the energy axis.

**Fig. 5 Packing of cluster polymer chains in the monoclinic unit cell.** **a** Packing inside a unit cell. The rods indicate the polymerization direction (*c*-axis). **b** Zoomed-up image of the marked area in **a**. The pink dots denote the centroids of the A-Adm groups. The average center-to-center distance and H···H contact distances between adamantyl moieties are 6.7 and 2.3 Å, respectively. Only weak van der Waals interactions are observed between the metallic chains along crystallographic *a*- or *b*-axes. Colors: golden and green, Au/Ag; gray, C. All hydrogen atoms are omitted for clarity.

be attributed to an increase of a single Ag–Au bond (viz. two in the monomer and three in the polymer as shown in Fig. 3, top, two Ag(b)–Au bonds of 2.86 Å (av) and bottom, three Ag(b)–Au bonds of 2.93 Å (av)). Of the three short Au···Au edges of the exterior (nonbonding) Au hexagon of the $(AuAg)_{31}L_{18}$ building block, only one is occupied by Ag(a) in the monomer but two are occupied by Ag(a) and Ag(b) (a and b are equivalent) in the polymer.

**Optical properties and electronic structure.** The solid-state optical diffuse reflectance spectra of $(AuAg)_{34}$ clusters both in monomeric and polymeric forms were measured (Fig. 6 and Supplementary Fig. 8). The spectra show two relatively weak peaks/shoulders in the range of 2.4–2.7 eV and a rather long tail towards the optical gap, which can be estimated from the extrapolated low-intensity absorptions as about 1.4 eV both for the monomeric $(AuAg)_{34}$ and the cluster polymer.

Optical properties and the electronic structure of the isolated $(AuAg)_{34}L_{20}$ cluster and the cluster polymer crystal were studied by extensive DFT computations (technical details discussed in Methods). Since the experimental crystal structure contained

positional Au/Ag disorder in 18 out of the 34 metal sites, we created four models for the isolated cluster: two models spanning both extremes of Au/Ag occupation ($Au_{19}Ag_{15}L_{20}$ and $Au_{26}Ag_8L_{20}$) and two models for the composition $Au_{21}Ag_{13}L_{20}$, which is close to the measured $Au_{21.3}Ag_{12.7}L_{20}$, and where the disordered sites were occupied randomly by gold or silver weighted by the experimental probability (Supplementary Table 2). We studied the electronic ground state structure of all these four model clusters by using the DFT implementation in the GPAW code[51] and computed the optical absorption spectra by using the linear-response (LR) formalism of the time-dependent DFT (LR-TDDFT)[52]. In all calculations, the electron–electron interactions were treated by using the Perdew–Burke–Ernzerhof (PBE) exchange-correlation potential[53], which we have found to be an acceptable compromise between accuracy and computational efficiency in numerous previous studies of ligand-stabilized noble metal clusters. The electronic structure was analyzed for both non-relaxed (all atom coordinates taken directly from the crystal data) and computationally relaxed clusters. The atomic coordinates of the DFT-relaxed models 1–4 described above are given as Supplementary Datasets 1–4.

The computed energy gaps between the highest occupied and lowest unoccupied molecular orbitals (HOMO-LUMO gaps) are consistently in the range of 1.31–1.40 eV for non-relaxed clusters and 1.28–1.46 eV for relaxed ones. LR-TDDFT calculations gave optical spectra that are compared to the experimental data in Supplementary Fig. 9. Comparing the spectra in energy scale one sees that all the four models yield absorption spectra that reproduce the shape of the experimental spectrum rather well up to about 3 eV, albeit a slight underestimation of the optical gap. Particularly one of the models where Au/Ag occupations were drawn randomly from the experimental distribution ("model 4") reproduces the full shape of the experimental spectrum extremely well up to 3 eV, including the two peaks/shoulders around 2.4–2.7 eV (comparison shown also in Fig. 6). Shapes of the spectra for relaxed clusters change slightly, most likely due to the fact that the PBE overestimates the metal–metal bonds by 2–4 %.

The non-relaxed cluster model 4 was then used to build the model for the cluster polymer crystal, with four clusters in the periodic crystal unit cell (Supplementary Fig. 10, and the coordinates given as Supplementary Dataset 5) and the electronic structure sampled by $4 \times 4 \times 4$ Monkhorst–Pack k-point mesh[54]

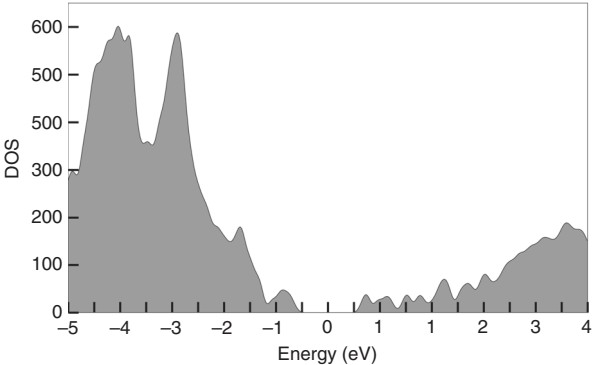

**Fig. 7 DFT-computed electronic density of states (DOS) of the cluster polymer crystal.** The cluster model 4 was used (as in Fig. 6) to build the periodic crystal, and the integration over the Brillion zone was done in a 4 × 4 × 4 Monkhorst–Pack k-point mesh. The band gap is centered around zero.

in the reciprocal space. The electronic density of states shown in Fig. 7 features broad valence ($E < 0$) and conduction ($E > 0$) bands with a fine structure. The apparent band gap, determined as the peak separation closest to the gap, is about 1.3 eV. This gap is comparable to the HOMO-LUMO and optical gap for the isolated cluster. The magnitude of the band gap indicates that the cluster polymer crystal should behave as a semiconductor-type material.

To gain qualitative insight into the characteristics of the valence and conduction bands, we formed a summed density from the occupied/unoccupied electron states forming the peaks closest to the band gap in Fig. 7 (top of the valence band at about −0.9 eV, and bottom of the conduction band at about +0.75 eV). This data is visualized in Supplementary Fig. 11 showing that appreciable density overlap between the neighboring clusters in the polymer exists only in the direction of the polymer axis. This implies that the electrical conductivity of the crystal should be anisotropic.

**Highly anisotropic *p*-type semiconductivity of the polymer crystal.** As Fig. 5 indicated, the **(AuAg)$_{34n}$** clusters form polymeric chains parallel to the *c*-axis of the single crystal, and these chains are separated in a- and b-directions by the bulky A-Adm ligands (Fig. 5b). This structural anisotropy together with the DFT-predicted electronic structure (discussed above) of the polymer crystal gives reasons to anticipate highly anisotropic (semi)conductivity.

To prove this, FET devices were fabricated to measure the direction-dependent conductivity of the polymer crystals (representative crystals shown in Supplementary Figs. 12 and 13) as discussed in Methods. The schematics of the single-crystal nanocluster FET chip are depicted in Fig. 8a. Detailed fabrication processes are described in Supplementary Figs. 14 and 15. In brief, four-terminal electrode sets (5 nm Cr/50 nm Au) were deposited on silicon wafer with 300 nm silicon dioxide using photolithography, followed by e-beam evaporation. Then a single crystal of cluster polymer was transferred onto the pre-patterned gold electrodes, with the device channel aligned with the a- or c-crystallographic axis, and by using silver paste to create the electrical contacts between the crystal and the electrode. The highly doped silicon wafer serves as a global backgate for the devices.

More than ten devices were studied in this work, each device comprising two orthogonal electrode pairs along the a- and c-crystallographic axes. The electrical conductivity was determined from the slope of the linear *I–V* curve. All the field-effect transistors showed the anisotropy of electrical conductivity, as

illustrated in Fig. 8b. The averaged electrical conductivity along the c-crystallographic axis of the crystal at room temperature and relative humidity of 56% is $1.49 \times 10^{-5}\,\mathrm{S\,m^{-1}}$, which is 1800 times of the electrical conductivity along the a-crystallographic axis (values averaged over six single crystals). Six measurements, plus the average and standard deviation are shown in Supplementary Table 5. Blank controls without crystal transferred was also measured, showing only instrument noise levels (Supplementary Fig. 16), which means that the conductivity is contributed by the crystal itself. Furthermore, the averaged conductivity of monomeric crystals is around $6.27 \times 10^{-8}\,\mathrm{S\,m^{-1}}$, near the conductivity along a-axis and much lower than the conductivity along c-axis of polymeric crystal (Supplementary Fig. 17). These results confirm that the direct linking of the clusters by the –Ag–Au–Ag– chains are beneficial for carrier transport.

We then probed the semiconductor properties of single crystal along c-crystallographic axis. The transfer and output characteristic curves are shown in Fig. 8c, d. The transfer curves show that at negative gate voltage ($V_\mathrm{G}$), the source-drain currents increase with more negative $V_\mathrm{G}$, demonstrating *p*-type field effect. This indicates a hole conduction mechanism. The ON/OFF current ratio is around 4000 and the charge carrier mobility reaches $2.46 \times 10^{-2}\,\mathrm{cm^2\,V^{-1}\,s^{-1}}$ at source-drain voltage ($V_\mathrm{SD}$) = −16 V calculated using the standard method in the unsaturation regime[55]. The exponential behavior in the output curves can be attributed to Schottky barrier between the electrode and the polymer crystal. Moreover, the transistor characteristic is reproducible as shown in the supporting information (Supplementary Fig. 18). As a comparison, recently reported photoconductive two-dimensional (2D) films of phosphine-thiolate-stabilized Au$_{25}$ clusters[36] showed ON/OFF ratio of about 50 000 for $V_\mathrm{SD} = 6$ V, charge carrier mobility approaching $10^{-5}\,\mathrm{cm^2\,V^{-1}\,s^{-1}}$ at $V_\mathrm{SD} = 20$ V, and n-type field effect. The mobility of our polymeric crystal of nanoclusters is in the range of traditional p-type single-crystal organic semiconductors[56] and close to the mobility of supercrystal of CdSe Quantum Dots (Supplementary Table 6)[57,58]. We further note that the conductivity ($1.49 \times 10^{-5}\,\mathrm{S\,m^{-1}}$, see above) of our crystals in the c-crystallographic axis is one to three orders of magnitude higher than the values reported recently for thiolate-stabilized 1D assemblies of Au$_{21}$ clusters[26] where 1D "nanofibrils" of the clusters were formed by modulating the weak interactions between the ligand layers of the clusters. These comparisons indicate that the conductivity and charge carrier mobility is increased by several orders of magnitude in our macroscopic cluster-based materials via direct linking of the clusters by the –Ag–Au–Ag– chains in the cluster polymer crystal.

## Discussion

We have demonstrated that the use of bulky adamantane alkyne facilitates the solvent-mediated self-assembly of atomically precise intermetallic **(AuAg)$_{34}$** nanoclusters into 1D cluster polymers that grow into macroscopic crystals with up to hundreds of microns in length, making the single crystals readily available for materials characterizations. The materials have highly anisotropic structure where the neighboring clusters are directly connected by –Ag–Au–Ag– linkages in the c-direction of the crystals but have insulating adamantane layers between the polymeric chains. We have tested the performance of this material as a component of a FET device and found that the device properties show about 1800-fold anisotropy in the conductance properties along and across the polymer chain, ON/OFF ratio of about 4000, p-type field effect, and hole carrier mobility of up to $2.46 \times 10^{-2}\,\mathrm{cm^2\,V^{-1}\,s^{-1}}$. This work offers a new insight into the structural factors for controlling the electron transport in assemblies of clusters and nanoparticles and

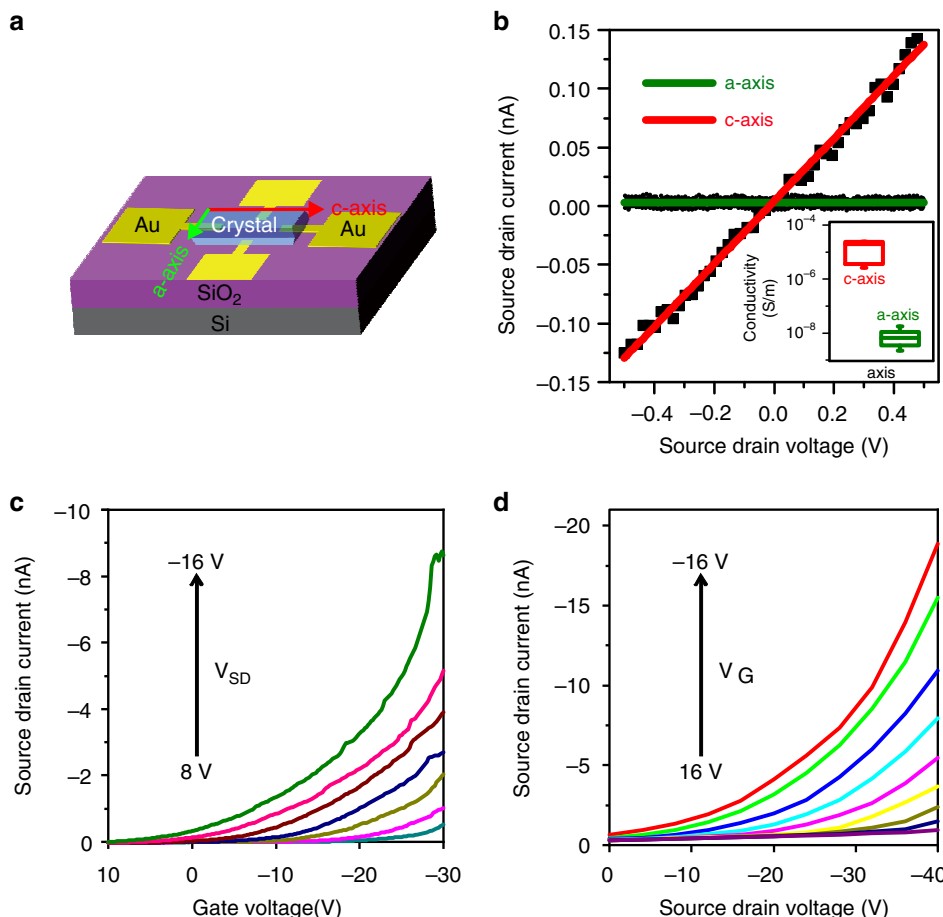

**Fig. 8 Electrical transport properties of the cluster polymer crystals. a** The structure of the polymer crystal FET. **b** *I–V* plot of the polymer crystal along *a*-axis and *c*-axis, respectively, with the range of corresponding conductivity values shown in the inset. **c** Transistor transfer ($V_{SD} = 8$ to –16 V in –4 V steps) and **d** output characteristics ($V_G = 16$ to –16 V in –4 V steps) measured along *c*-axis of the polymer crystal. Channel length, width = 80, 50 μm.

enhances our abilities to create new hierarchical nanoparticle assemblies with desired structures and properties.

## Methods

**Materials**. All materials were purchased from Alfa Aesar without further purification. Silver acetate (AgOAc, purity 98%), Borane-tert-butylamine complex [$(CH_3)_3CNH_2 \cdot BH_3$, purity 97%], 1-Ethynyladamantane ($C_{12}H_{16}$, purity 95%), Sodium methoxide ($CH_3ONa$, powder, purity 99%), Potassium methoxide ($CH_3OK$, powder, purity 99%), dichloromethane ($CH_2Cl_2$, analytical grade) and methanol ($CH_3OH$, analytical grade) were purchased from Sinopharm Chemical Reagent Co. Ltd. (Shanghai, China). Leitsilber 200 Silver Paint was puchased from Tedpella. All reagents were used as received without further purification. The water used in all experiments was ultrapure. All reagents were used as received without further purification. AuSMe$_2$Cl was prepared according to literature methods[59].

**Physical measurements**. UV–Vis absorption spectra were recorded on a Varian Cary 5000 spectrophotometer. Energy-dispersive X-ray spectroscopy (EDS) was performed on a scanning electron microscopy (SEM, Hitachi S4800) equipped with energy-dispersive X-ray spectroscopy operated at 15 kV. The compositions of **(AuAg)$_{34n}$** and **(AuAg)$_{34}$** were determined by inductively coupled plasma mass spectrometry (ICP-MS, Agilent 7700).

**Single-crystal analysis**. The diffraction data were collected on an Agilent SuperNova X-Ray single-crystal diffractometer using Cu Kα ($\lambda = 1.54184$ Å) and Mo Kα ($\lambda = 0.71073$ Å) micro-focus X-ray sources at 100 K. The data were processed using CrysAlisPro. The structure was solved and refined using Full-matrix least-squares based on $F^2$ with programs SHELXT and SHELXL[60] within OLEX2[61].

**Synthesis of (AuAg)$_{34n}$**. Six milligrams of AuSMe$_2$Cl was first dissolved in the mixture solution of dichloromethane and methanol. In all, 3.5 mg 1-Ethynyladamantane and 5 mg sodium methoxide were then added to the solution and stirred for 20 min. Five milligrams of AgCH$_3$COO was added and stirred for

5 min. The reducing agent, tert-butylamineborane (3.6 mg), was added to the mixture under vigorous stirring. The reaction was aged for 12 h at room temperature. The solution was centrifuged for 4 min at 10,000 r min$^{-1}$. The brown supernatant was subjected to natural evaporation in the dark. Red block crystals were obtained within 1 week in a yield of ~35% (based on Au).

**Synthesis of (AuAg)$_{34}$**. Six milligrams of AuSMe$_2$Cl was first dissolved in the mixture solution of chloroform and methanol. In alll, 3.5 mg 1-Ethynyladamantane and 5 mg sodium methoxide were then added to the solution and stirred for 20 min. Five millihrams of AgCH$_3$COO was added and stirred for 5 min. The reducing agent, tert-butylamineborane (3.6 mg), was added to the mixture under vigorous stirring. The reaction was aged for 12 h at room temperature. The solution was centrifuged for 4 min at 10,000 r min$^{-1}$. The brown supernatant was subjected to natural evaporation in the dark. Red block crystals were obtained within 1 week in a yield of ~25% (based on Au).

**DFT computations**. The density functional theory (DFT) computations were obtained with the GGA-PBE exchange-correlation functional[53] employing a uniform real-space grid for representing the wave functions as coded in the GPAW software[51]. The grid-spacing was 0.2 Å for all the calculations. The optical spectra for the monomers were calculated with the linear-response time-dependent DFT (LR-TDDFT) as implemented in GPAW[52].

The effect of the ratio of gold to silver atoms was tested using four different compositions of the metal core for the monomers: Au$_{19}$Ag$_{15}$, Au$_{26}$Ag$_8$ and two different isomers of Au$_{21}$Ag$_{13}$. The atomic coordinates with fractional occupancies were assigned employing the probabilities from Supplementary Fig. 1. For Au$_{19}$Ag$_{15}$, the element with the larger probability was always chosen (model 1). For Au$_{26}$Ag$_8$, the opposite choice was made (model 2). The metal atoms for the trial clusters with a core of Au$_{21}$Ag$_{13}$ were randomly assigned using the probabilities as weights (model 3, 4). The calculation box was chosen for each system so that 6 Å of vacuum was left on each side. All four structures were also relaxed, requiring the final structure to have forces of <0.05 eV Å$^{-1}$ on each atom.

For all the monomers, the UV–Vis spectrum was calculated for the coordinates taken directly from the crystal structure, as well as the relaxed structure, and these

spectra were compared to the measured one. The electron-hole excitations with an energy up to 4.75 eV (≈260 nm) were included in the calculations. A Gaussian broadening of 0.1 eV for individual oscillator strengths was used to plot the spectra.

For the polymer, the model cluster 4 was chosen, since it had the best fit between the experimental and computational UV–Vis spectrum for the monomer. A calculation with periodic boundary conditions was performed using the positions and the unit cell determined from the crystal structure. The monoclinic unit cell has four clusters inside. Monkhorst–Pack sampling[54] with 4 x 4 x 4 k-points was used for the Brillouin-zone. The total density of states (DOS) was calculated using a Gaussian broadening of 0.1 eV.

**Fabrication of single-crystal nanoclusters-based transistors**. The fabrication procedure is illustrated in Supplementary Fig. 14. First, a layer of photoresist (AZ5214E) was spin-coated on 300 nm $SiO_2$ covered Si wafer and metal electrodes (5 nm Cr/50 nm Au) were patterned by photolithography, followed with electron beam evaporation and lift-off process. Then a single crystal of cluster polymers was carefully transferred onto contact with the pre-patterned electrodes, served as source-drain electrode. The heavily doped silicon wafer serves as backgate electrode. Conductive silver epoxy was introduced to glue them ensuring a better contact. After the conductive silver epoxy was completely dry for about 2 h, we could conduct the electrical properties measurement.

**Electrical properties measurement**. The electrical conductivity of single crystal along different crystallographic axes was measured at room temperature at relative humidity of 56% using a Keithley 4200-SCS source meter and a MPS150 manual probe station.

## Data availability

The X-ray crystallographic coordinates for structures reported in this work have been deposited at the Cambridge Crystallographic Data Center (CCDC), under deposition numbers 1962411 and 1962412 for the polymeric $(AuAg)_{34n}$ and $(AuAg)_{34}$ clusters, respectively. These data can be obtained free of charge from the CCDC via www.ccdc.cam.ac.uk/data_request/cif. The coordinate files for the DFT-optimized cluster models 1-4 and the coordinate file for the polymer crystal used in the DFT calculation are given as part of the electronic supplementary material (Supplementary Datasets 1–5). Checkcif files for $(AuAg)_{34}$ cluster and $(AuAg)_{34n}$ polymer CIF files are given as Supplementary Dataset 6 and 7, respectively. All other data discussed in this work is available by reasonable requests to the corresponding authors.

## Code availability

The DFT code used in this work is publicly available and documented at https://wiki.fysik.dtu.dk/gpaw/.

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

## Acknowledgements

The experimental work was supported by the National Key R&D Program of China (2017YFA0207302, 2018YFA0306900 and 2018YFA0209500), the National Natural Science Foundation of China (21890752, 21731005, 21721001 and 21872114) and the fundamental research funds for central universities (20720180026). The computational work was supported by the Academy of Finland (grants 294217, 319208, and H.H.'s Academy Professorship). N.F.Z. acknowledges the support from the Tencent Foundation through the XPLORER PRIZE. H.H. acknowledges support from China's National Innovation and Intelligence Introduction Base visitor program. E.S. acknowledges Emil Aaltonen Foundation for a Ph.D study grant and thanks O. Lopez-Estrada for technical help in setting up the DFT calculations. All the computations were done at the Barcelona Supercomputing Center under a PRACE computing grant 2018194723.

## Author contributions

P.Y. performed the synthesis, crystallization, and single-crystal X-ray analysis of the materials and analyzed the data with B.K.T. R.H.Z. performed the fabrication of FET devices and conductance measurements. P.P.R. assisted the synthesis of organic ligands. Y.R.Y. assisted the X-ray structure analysis. Y.Z. carried out the measurement of ICP-MS. N.F.Z. and Y.C. supervised the experimental work. E.S. performed the DFT calculations and analyzed data together with S.M. H.H. supervised the computational work. The manuscript was edited by H.H., B.K.T., Y.C., and N.F.Z.

## Competing interests

The authors declare no competing interests.
