## [Peer Review File · Nature Communications]

Reviewers' comments:

Reviewer #1 (Remarks to the Author):

Rational design of bulky device based on nanosized nanomaterials has long been an important research direction in material science. Diverse methods have been developed to control the self-assembly behavior of nanoparticles in order to control the properties of final device. These methods based on traditional nanoparticles are faced with two problems. One is the control on the properties of single nanoparticle is nearly impossible as the structure of nanoparticles cannot reach atomic precision. The other one is the assembly of these nanoparticles cannot be precisely controlled. Though there are some factors like oriented-attachment can control the direction and contact between neighbored nanoparticles, the defects in the device cannot be avoided yet.

It's quite exciting to see that these two problems can be solved by using atomically-precise metal nanoclusters (which could also be realized via total synthesis pathway). In this manuscript, the authors showed that: 1) the building block of the bulky device is atomically precise, a (AuAg)₃₄ nanocluster; 2) the connection between neighbored nanoclusters is atomically precise, Ag-Au-Ag bonds. Importantly, this kind of atomic precision can be realized in bulky device as the structure of supercrystal composed of 1D polymers is highly ordered, high enough to be resolved by X-ray diffraction analysis. I can almost foresee many new research works in this direction. Therefore, I would like to recommend its publication in Nature Communication after the authors address the following minor issues.

1. The structure of the cluster monomer and the 1D polymer is interesting. It seems that the AuAg₂ acts as an excess part that adsorbed on the core of the nanocluster, and at the same time acts as the linker in the polymerization process of the monomers. Does the (AuAg)₃₁ nanocluster exists? Is it possible to synthesized the main block, like (AuAg)₃₁ in this case, and then introduce crosslinker, like AuAg₂, to build cluster polymers?
2. Does there any new molecular orbit formed in the polymerization process? What about the differences between the absorption spectra of supercrystal of monomers and the 1D polymer?
3. You mentioned in the synthesis of the polymer that the super crystal of monomers can be converted to 1D polymer supercrystal. Does this process overcome a re-dissolve and crystallization process? Or the super crystal can transform in-situ?
4. Can you list the carrier mobilities of some traditional p-type semiconductors or that of devices composed of super crystal of nanoparticles of p-type semiconductors? We then can find out what's the position of device made of atomically-precise metal nanoclusters.

Reviewer #2 (Remarks to the Author):

Häkkinen and coworkers present a synthesis of crystalline 1D wires comprising adamantane passivated AuAg clusters. The synthesis is simple and straightforward, although it does not appear to be rational. The use of methylene chloride is important for forming extended structures, but for reasons not understood. However, these structures are novel, as there are only a few other 1D chains known in the literature containing AuAg clusters.

The single-crystal device studies are well done and demonstrate that these materials are anisotropic conductors as well as p-type semiconductors. Overall I believe the work is novel and should be published in Nature Communications.

I have a few questions regarding the characterization:

1. Do these polymeric structures exist in solution? An optical absorption study in chloroform would be nice to see. The authors mention ICP-MS but only present data in the ESI in table format.
2. The optical absorption studies presented are in the solid state. What about diffuse reflectance? This would allow for a Tauc plot to be constructed to determine a more accurate band gap.
3. The conductivity is tested using a two point probe method, and 10 devices are reportedly tested, which I applaud the authors for reporting. Is the reported value for the "champion" device or an average of all 10 devices?
4. Is it possible for the authors to report a 4-point probe measurement for their system? This is recommended for measuring electrical conductivity in metal-organic framework materials. See: J. Am. Chem. Soc. 2016, 138, 44, 14772-14782

Reviewer #3 (Remarks to the Author):

In this manuscript, the authors reported a solvent-mediated assembly of 34-atom intermetallic gold-silver clusters protected by twenty 1-ethynyladamantanes into 1D polymers with Ag-Au-Ag bonds between neighboring clusters. They also utilized the anisotropic property in the structure to fabricate an FET, demonstrating an anisotropic conductivity. Several comments as below are required to be considered.

1. Why do you use the word 'atom-precise' in the title?
2. Many references [22-38] are cited in the introduction. It is better to elaborate on various applications of assemblies of gold, silver and copper clusters in biochemical sensing, optoelectronics, and catalysis.
3. In the legend of Figure 1, '(b)' is needless.
4. In Figure 2, why do you analyze (AuAg)₃₄ clusters from a smaller cluster (AuAg)₁₃? Is it a fundamental

cluster to compose various assemblies of AgAu clusters?

5. Please try to explain the different roles of CHCl_3 and CH_2Cl_2 in the polymerization reaction of $(\text{AgAu})_{34}$ clusters, and the mechanism beneath.

6. In consideration of the significance of HOMO and LUMO orbits in determining various physicochemical properties, please analyze the detailed compositions of HOMO and LUMO orbits since you have performed the calculations of electronic structures.

7. It is better to compare the electrical property between the monomeric crystal and the polymeric crystal.

8. Hall effect is better to be measured to further confirm the p-type character of the polymer crystal.

9. Please try to correlate the p-type character with the unique structure of the polymer crystal.

Reviewer #4 (Remarks to the Author):

The authors present the crystal structure of an atomically precise nanoparticle, which forms both a monomer and a polymer depending on conditions. There are not yet many nanoparticles that form polymer-like structures, so it is novel that way. Most of the article is a discussion of the crystal structure geometry, with less on the semiconducting information. The article is interesting, but I am not sure it is time-sensitive to publish in Nature Communications.

The .cif file for the monomer is very hard to visualize using Mercury. There are a number of issues regarding disorder in the crystal structure (some of which the authors discuss in their checkcif file), but it means that it is hard for the reader to view the structure of this nanoparticle in the paper. There are some places where either a Au or Ag atom may be present (it is nice that the percentages are given in Supplementary Table 1). What is the disorder that is responsible for the lack of bonds drawn in the .cif file? I looked in this file to try to see what types of interactions might be present between $(\text{AuAg})_{24}$ monomers in that crystal structure, but was unable to do this. What are the nearest intercluster Au/Ag - Au/Ag distances in the monomer crystal structure?

The Ag-Au-Ag linkage in the polymer is interesting. There is certainly some disorder... according to the .cif file, there appear to be 5 places where the Au atom can appear. However, the associated ligands are not so disordered. I can see how the outer 4 Au atoms can interact... what about the central Au position? This strikes me as a bit strange considering the other 4 positions that the Au atom can take.

The percentages of Au/Ag in certain sites seem to be very specific (close to, but not exactly equal to, 1/3 and 2/3). (Supp Table 1) Could the authors explain this? Are there any patterns to the Au-Ag positions?

It would be helpful if the atom descriptions given in Supplementary Tables 1 and 2 were accompanied by a figure that used arrows (or similar) to match the descriptions and the positions. This would be especially helpful because the terms used here are not the same as those on p. 4 of the text. I think I have been able to figure out some or all of the positions, but I am not confident in my assignments in

relation to the text.

Supplementary Figure 4 has parts listed in the figure, but the caption does not have corresponding letters. The caption seems to refer to two structures, whereas three parts of the figure are given.

Is the monomer to polymer transformation reversible (either by using a different solvent or by heating)?

The coordinates of relaxed model clusters 1-4 and the polymer should be provided in the SI.

Would the authors have information about the occupied/virtual or valence/conduction band character in this system?

It is good that the authors provided the average electrical conductivity over 6 single crystals. It would be best if they present the average and the standard deviation for the conductivity along the c and a directions (a table in the Supporting Information would be fine) (or perhaps include all six measurements, plus the average and standard deviation) so that readers can know about the spread of the values measured.

A reference should be included for "AuSMe₂Cl was prepared according to literature methods". p. 12

Reply to Referee # 1

We thank the Referee for his/her insightful comments that helped us to make the paper clearer and stronger.

1. Comment: The structure of the cluster monomer and the 1D polymer is interesting. It seems that the AuAg₂ acts as an excess part that adsorbed on the core of the nanocluster, and at the same time acts as the linker in the polymerization process of the monomers. Does the (AuAg)₃₁ nanocluster exist? Is it possible to synthesize the main block, like (AuAg)₃₁ in this case, and then introduce crosslinker, like AuAg₂, to build cluster polymers?

Response: The crystal structure analyses reveal that (AuAg)₃₄ and (AuAg)_{34n} nanoclusters have the same block, (AuAg)₃₁(Adm)₂₀. The AuAg₂ unit acts as a linker to connect the (AuAg)₃₁ building blocks into polymers in the polymerization process. Indeed, we have tried to use MALDI-TOF to track the crystallization process of (AuAg)_{34n} nanoclusters. Interestingly, we got two strong peaks at 7866 and 8955 under positive-ion mode (Figure R1). These two peaks can be attributed to [Au_{20.4}Ag_{10.6}(Adm)₁₇]⁺ and [Au_{22.4}Ag_{12.6}(Adm)₂₁]⁺, respectively (Figure R2 and R3). It seems that the polymeric nanocluster [Au_{21.4}Ag_{12.6}(Adm)₁₇]_n is made of two different components. We also tried different crystallization methods to capture the (AuAg)₃₁ block. Unfortunately, we always got the polymeric (AuAg)_{34n} nanoclusters. Using cross-linkers to link premade nanoclusters is a very promising strategy to prepare polymers of metal nanoclusters, which should be extensively investigated in our future research.

Figure R1. MALDI-TOF spectrum of the crude product of (AuAg)_{34n} nanocluster.

Figure R2. The possible structure of $\text{Au}_{20.4}\text{Ag}_{10.6}(\text{Adm})_{18}$. Colors: golden and green, Au/Ag; grey, C. All hydrogen atoms are omitted for clarity.

Figure R3. The possible structure of $\text{Au}_{22.4}\text{Ag}_{12.6}(\text{Adm})_{22}$. Colors: golden and green, Au/Ag; grey, C. All hydrogen atoms are omitted for clarity.

2. Comment: Does there any new molecular orbit formed in the polymerization process? What about the differences between the absorption spectra of supercrystal of monomers and the 1D polymer?

Response: Our DFT analysis showed that no new molecular orbitals formed in the polymerization process close to the polymer band gap. The electronic states that have the highest weight around the linker Au atom are centered well below the top of the valence band (not shown). The UV-vis absorption spectrum of the polymeric and the monomeric in solid exhibit two relatively weak absorption peaks at ~ 420 , and ~ 515 nm (Supplementary Figure 7). Compared to the monomeric $(\text{AuAg})_{34}$ nanocluster, the peaks of the polymeric $(\text{AuAg})_{34}$ were red shifted, suggesting a slightly smaller energy

gap of the polymer than that of the monomeric cluster. The UV–vis spectra of $(\text{AuAg})_{34n}$ and $(\text{AuAg})_{34}$ in solid have been added to the Supplementary Information (Supplementary Figure 7).

3. Comment: You mentioned in the synthesis of the polymer that the super crystal of monomers can be converted to 1D polymer supercrystal. Does this process overcome a re-dissolve and crystallization process? Or the super crystal can transform in-situ?

Response: It is interesting to note that the monomeric nanocluster can be converted to 1D polymeric nanocluster in solution. When the crystals of the monomeric $(\text{AuAg})_{34}$ were re-dissolved in a mixture of methanol and CH_2Cl_2 , crystals containing the cluster polymers were obtained. We agree with the reviewer that this process should overcome a re-dissolving and crystallization process.

$(\text{AuAg})_{34}$ and $(\text{AuAg})_{34n}$ share the same $(\text{AuAg})_{31}\text{L}_{18}$ unit. The Ag-Au-Ag unit is bent in the monomeric $(\text{AuAg})_{34}$ but linear in the polymeric $(\text{AuAg})_{34n}$, which is the key to the structural transformation from $(\text{AuAg})_{34}$ to $(\text{AuAg})_{34n}$. We failed to realize this transform process in-situ, which might be due to the difficulty to transform the Ag-Au-Ag unit from bent to linear structure.

4. Comment: Can you list the carrier mobilities of some traditional p-type semiconductors or that of devices composed of super crystal of nanoparticles of p-type semiconductors? We then can find out what's the position of device made of atomically-precise metal nanoclusters.

Response: We have listed the mobilities of traditional p-type semiconductors and super crystal of nanoparticles of semiconductors as suggested (Supplementary Table 5) in the revised manuscript. We found a few references reporting electron mobilities of n-type but found no references of p-type supercrystal of nanoparticles. The mobility of our polymer crystal of nanoclusters is $0.02 \text{ cm}^2 \text{ V}^{-1} \text{ s}^{-1}$, within range of traditional p-type single-crystal organic semiconductors and close to the mobility of supercrystal of CdSe quantum dots. The following result discussion has been added to the main text: “The mobility of our polymeric crystal of nanoclusters is in the range of traditional p-type single-crystal organic semiconductor⁵⁶ and close to the mobility of supercrystal of CdSe quantum dots (Supplementary Table 5).^{57,58}”

Reply to Referee # 2

We thank the Referee for his/her insightful comments that helped us to make the paper clearer and stronger.

1. Comment: Do these polymeric structures exist in solution? An optical absorption study in chloroform would be nice to see. The authors mention ICP-MS but only present data in the ESI in table format.

Response: The polymeric structure cannot be dissolved in solvents such as chloroform. So we were not able to measure the absorption spectrum in solutions. At this point, we should not exclude the possible presence of oligomers or clusters in the solution. As we replied to Referee 1, we used MALDI-TOF to track the crystallization process of $(\text{AuAg})_{34n}$ nanocluster. Interestingly, two strong peaks at 7866 and 8955 under positive-ion mode (Figure R1) were observed. These two peaks can be attributed to $[\text{Au}_{20.4}\text{Ag}_{10.6}(\text{Adm})_{17}]^+$ and $[\text{Au}_{22.4}\text{Ag}_{12.6}(\text{Adm})_{21}]^+$, respectively (Figure R2 and R3). Together with these two strong peaks, two weak peaks at $\sim 1.1\text{k}$ and $\sim 1.5\text{k}$ were also revealed. The peak of $\sim 1.5\text{k}$ can be attributed to the dimers of $\text{Au}_{21.4}\text{Ag}_{12.6}(\text{Adm})_{20}$ nanocluster. The MS data indicate that these oligomeric species of the nanoclusters might exist in solution. In the revised manuscript, we have included the description of the ICP-MS measurement as follows: “The X-ray results are confirmed by independent compositional assignments made using of ICP-MS (1:1.69 and 1:1.73 molar ratio of Ag/Au for total 34 metallic atoms in $(\text{AuAg})_{34}$ and $(\text{AuAg})_{34n}$ respectively) and EDS (1:1.67 and 1:1.74 molar ratio of Ag/Au for total 34 metallic atoms in $(\text{AuAg})_{34}$ and $(\text{AuAg})_{34n}$ respectively) (Supplementary Figs. 2, 3, and Supplementary Table 3).”

2. Comment: The optical absorption studies presented are in the solid state. What about diffuse reflectance? This would allow for a Tauc plot to be constructed to determine a more accurate band gap.

Response: In fact, the solid-state UV–vis absorption spectra of $(\text{AuAg})_{34}$ clusters both in monomeric and polymeric forms were measured in the mode of diffuse reflectance. In the revised manuscript, the following result discussion has been added to the main text: “The solid-state optical diffuse reflectance spectra of $(\text{AuAg})_{34}$ clusters both in monomeric and polymeric forms were measured (Supplementary Figure 7).” The UV–vis spectra of $(\text{AuAg})_{34n}$ and $(\text{AuAg})_{34}$ in solid have also been added to the Supplementary Information (Supplementary Figure 7).

3. Comment: The conductivity is tested using a two point probe method, and 10 devices are reportedly tested, which I applaud the authors for reporting. Is the reported value for the "champion" device or an average of all 10 devices?

Response: Thank you for the comment. We have fabricated more than 10 devices. The reported value is not for the "champion" device. Actually, we report the average value of 6 devices with conductivities in a narrow range. We have now revised the manuscript to make it clear. All six measurements, plus the average and standard deviation are newly added in the supporting information (Supplementary Table 4). In the revised manuscript, the following result discussion has been added to the main text: "The averaged electrical conductivity along the c-crystallographic axiswhich is 1800 times of the electrical conductivity along the a-crystallographic axis (values averaged over 6 single crystals). All six measurements, plus the average and standard deviation are shown in Supplementary Table 4."

4. Comment: Is it possible for the authors to report a 4-point probe measurement for their system? This is recommended for measuring electrical conductivity in metal-organic framework materials. See: J. Am. Chem. Soc. 2016, 138, 44, 14772-14782

Response: The 4-point probe measurement is ideal for accurate measurements of electrical conductivity. We had tried several methods, including top-contact, bottom-contact and conductive paste methods (J. Am. Chem. Soc. 2016, 138, 44, 14772) to perform 4-point probe measurement but without success. For the top-contact method, it is incompatible with our 50 μ m thick single crystal because the evaporated metallic thin film is usually less than 1 μ m. The bottom-contact method, only contacting one crystal surface, makes it difficult to measure the cross-sectional area of the conducting channel. Moreover, due to the small size (length less than 200 μ m) and brittleness of our crystals, it is challenging to fabricate 4-contact onto a 200 μ m-long crystal without short-circuiting the leads by using conductive paste method. The difficulty is also encountered by the research you mentioned. "*Due to the small size of single crystals of MOFs (<1 mm), methods involving four contacts can be challenging*". They showed a 4-point probe measurement with ~2 mm crystal. If larger crystals can be grown, the 4-point probe measurement would be feasible. At the same time, the authors in this

reference also pointed out that “*the typical resistance of wires and contacts is less than 100 Ω . Therefore, to measure electrical conductivity with less than 10% error, the resistance of the sample needs to be higher than 1 k Ω* ”. Since the resistance of our crystal is about giga-ohm, the resistance we measured should mainly be contributed by the crystal itself.

Reply to Referee # 3

We thank the Referee for his/her insightful comments that helped us to make the paper clearer and stronger.

1. Comment: Why do you use the word ‘atom-precise’ in the title?

Response: The term is used because the nanoclusters investigated in this work have atomically precise structures with well-defined atoms’ coordinates. Nanoclusters represent ideal models to resolve various important issues related to surface science and significantly expand our understanding of the structure-property correlation of novel functional metal nanomaterials.

2. Comment: Many references [22-38] are cited in the introduction. It is better to elaborate on various applications of assemblies of gold, silver and copper clusters in biochemical sensing, optoelectronics, and catalysis.

Response: As suggested, we have elaborated the references which are cited in the introduction. In the revised manuscript, the following introduction has been added to the main text: “This has inspired efforts towards nanoscale and macroscale assemblies of gold,^{22-23,33-35} silver,²⁵⁻²⁶ and copper clusters²⁹⁻³² that might have a variety of applications in the fields of (bio)chemical sensing,^{24,27-28} and optoelectronics.^{29-32,36-37}”

3. Comment: In the legend of Figure 1, ‘(b)’ is needless.

Response: Thanks for pointing out this mistake. In the revised manuscript, we have deleted the sentence “(b) The same 31-atom metal core structure of (AuAg)₃₄ and (AuAg)_{34n} nanoclusters.”

4. Comment: In Figure 2, why do you analyze (AuAg)₃₄ clusters from a smaller cluster (AuAg)₁₃? Is it a fundamental cluster to compose various assemblies of AgAu clusters?

Response: We describe the structures of (AuAg)₃₄ and (AuAg)_{34n} nanoclusters from (AuAg)₁₃ mainly because the centered icosahedral M₁₃ structure has been well-documented as cores for many metal clusters in the field of cluster chemistry. Starting from M₁₃ would help readers understand the total structure better. But at this point, we should not conclude that the M₁₃ nanoclusters are synthetically the building blocks or intermediates of the obtained (AuAg)₃₄ and (AuAg)_{34n} nanoclusters.

5. Comment: Please try to explain the different roles of CHCl_3 and CH_2Cl_2 in the polymerization reaction of $(\text{AgAu})_{34}$ clusters, and the mechanism beneath.

Response: In the literature, solvent has been well recognized as a crucial factor in the synthesis of metal nanoclusters. The stability and surface activity of metal nanoclusters are critically dependent on the solvent. For example, the structural transition of Au nanoclusters were readily achieved by a solvent-exchange method (J. Am. Chem. Soc., 2012, 134, 17997). CHCl_3 and CH_2Cl_2 have different physicochemical properties (e.g. polarity, viscosity) which might change the structure of surface ligands on metal nanoclusters during the synthesis and crystallization process. Another possible mechanism is the presence of different amount of free Cl^- species in CHCl_3 and CH_2Cl_2 . It has been well-demonstrated that Cl^- released from CHCl_3 and CH_2Cl_2 can heavily influence the formation of nanoclusters with the coordination of Cl^- to metal precursors as well as onto the surface of nanoclusters.

6. Comment: In consideration of the significance of HOMO and LUMO orbits in determining various physicochemical properties, please analyze the detailed compositions of HOMO and LUMO orbits since you have performed the calculations of electronic structures.

Response: We have looked at the symmetries of HOMO and LUMO orbitals, but in this case, it is more instructive to look at a group of close-lying orbitals near the bandgap in the polymer crystal. We have looked at the density of states of a few highest occupied states / unoccupied states (forming the upper edge of the valence band / lower edge of conduction band and the small DOS peaks below/above the bandgap in Fig. 7). The visualizations of the summed DOS of these states are shown in the new Supplementary Fig. 17. We see from the figure, that (i) appreciable DOS overlap between the neighbouring clusters in the polymer exists **only** in the direction of the polymer axis, and (ii) the overlap seems to be slightly larger in the case of the upper edge of the valence band (panel (a)).

(See also our response to Referee 3, comment 9; and Referee 4, comment 8)

7. Comment: It is better to compare the electrical property between the monomeric crystal and the polymeric crystal.

Response: We have followed the suggestion and measured electrical property of the

monomeric crystal. More than 5 devices of monomeric crystals were measured. The averaged conductivities of monomeric crystals is around $6.27 \times 10^{-8} \text{ S m}^{-1}$, which is similar to the conductivity along a-axis ($8.29 \times 10^{-9} \text{ S m}^{-1}$) and much lower than the conductivity along c-axis of polymeric crystal ($1.49 \times 10^{-5} \text{ S m}^{-1}$) (Supplementary Figure 15). This result again confirms that the direct linking of the clusters by the -Ag-Au-Ag- chains are beneficial for carrier transport. The following discussion has been added to the main text of our revised manuscript: “Furthermore, the averaged conductivity of monomeric crystals is around $6.27 \times 10^{-8} \text{ S m}^{-1}$, near the conductivity along a-axis and much lower than the conductivity along c-axis of polymeric crystal (Supplementary Figure 15). These results confirm that the direct linking of the clusters by the -Ag-Au-Ag- chains are beneficial for carrier transport.”.

8. Comment: Hall effect is better to be measured to further confirm the p-type character of the polymer crystal.

Response: Thank the reviewer for the suggestion. Hall effect can indeed be used to determine whether the semiconductor is n-type or p-type. However, the value of Hall voltage we calculated is less than 0.2 mV, much lower than Johnson noise. Considering this issue, it is challenging to determine the semiconductor type of our polymer crystal by measuring Hall effect.

The processes of calculation are shown as follows:

Considering the balance of Lorentz force and electric force on the charge carriers,

$$q\upsilon B = qU_H/L_a$$

We see that,

$$U_H = B\upsilon L_a$$

In this formula, the width of the crystal, L_a , is 50 μm . And the magnetic field strength B is set as large as 10 T (the maximum strength in our setup). While the velocity of the carrier, υ , can be determined from mobility μ and electrical field E:

$$\upsilon = \mu E = \mu V_{SD}/L = 0.02 \text{ cm}^2 \text{ V}^{-1} \text{ s}^{-1} \times 16 \text{ V}/80 \mu\text{m} = 0.4 \text{ m s}^{-1}$$

So the Hall voltage can be calculated as

$$U_H = BvL_a = 10 \text{ T} \times 0.4 \text{ m s}^{-1} \times 0.00005 \text{ m} = 0.2 \text{ mV}.$$

9. Comment: Please try to correlate the p-type character with the unique structure of the polymer crystal.

Response: Thank the reviewer for the insightful suggestion. The ligand of our polymer crystal, 1-ethynyladamantane, is deprotonated to form a monovalent anion. Thus, according to the composition of $(\text{AuAg})_{34}\text{L}_{20}$, we can presume that twenty Au/Ag atoms are positively charged with monovalent, while fourteen Au/Ag atoms will be zero-valent. Owing to the existence of zero-valent metal as well as the anionic ligand, we believe that our polymer crystal is electron rich, which is the character of organic p-type semiconductor (Ref: Different Types Field-Effect Transistors-Theory Applications. 2017). Additional insight can be gained from the DFT analysis of the polymer crystal. The sum of densities of a few occupied and unoccupied electronic states around the band gap, shown in newly added Supplementary Fig. 17, indicates that overlapping state density between neighbouring clusters in the direction of the polymer for the occupied states is larger than in the unoccupied states. Thus, we believe that the charge transport through the valence band is more favored, more specifically, the hole conductivity in the top of valence band should be favored.

Reply to Referee # 4

We thank the Referee for his/her insightful comments that helped us to make the paper clearer and stronger.

1. Comment: The .cif file for the monomer is very hard to visualize using Mercury. There are a number of issues regarding disorder in the crystal structure (some of which the authors discuss in their checkcif file), but it means that it is hard for the reader to view the structure of this nanoparticle in the paper. There are some places where either a Au or Ag atom may be present (it is nice that the percentages are given in Supplementary Table 1). What is the disorder that is responsible for the lack of bonds drawn in the .cif file? I looked in this file to try to see what types of interactions might be present between (AuAg)₂₄ monomers in that crystal structure, but was unable to do this. What are the nearest intercluster Au/Ag - Au/Ag distances in the monomer crystal structure?

Response: The disorder in the Ag-Au-Ag unit in the monomeric structure is caused by the symmetry of the crystal structure. As shown in Supplementary Figure 5, there are three possibilities of the nearest adjacent nanoclusters in the real cell unit. For the three possibilities of the nearest adjacent nanoclusters, the nearest Au/Ag - Au/Ag distances should be 5.348, 6.814 and 10.241 Å respectively.

To make it easy for readers to view the structure of this monomeric nanocluster in this paper, together with Supplementary Figure 5, the following discussion has been added to the main text of the revised manuscript: “The Ag-Au-Ag unit in the monomeric nanocluster is disorder because of the symmetry of the crystal structure (Supplementary Fig. 5b). For the three possibilities of the nearest adjacent nanoclusters in the real unit cell, the nearest Au/Ag - Au/Ag distances should be 5.348, 6.814 and 10.241 Å respectively (Supplementary Fig. 5c, 5d and 5e).”

2. Comment: The Ag-Au-Ag linkage in the polymer is interesting. There is certainly some disorder... according to the .cif file, there appear to be 5 places where the Au atom can appear. However, the associated ligands are not so disordered. I can see how the outer 4 Au atoms can interact... what about the central Au position? This strikes me as a bit strange considering the other 4 positions that the Au atom can take.

Response: It is true that the Au atom in the Ag-Au-Ag linkage is disordered (Figure R4a). As shown in Figure R1c, there do exist 4 strong Q peaks around the Au atom. All of the Q peaks can be assigned to Au with the occupancies of 0.36, 0.29, 0.29, 0.04 and

0.04, respectively (Figure R4b). The 4 Au atoms are surrounding the central Au atom on a plane. The associated ligands are not so disordered mainly because of the π -type bonding with Ag.

Figure R4. (a) The dimers structure of $\text{Au}_{21.3}\text{Ag}_{12.7}(\text{Adm})_{20}$ nanocluster. (b) The Ag-Au-Ag linkage in the polymeric nanocluster. (c) The solution of The Ag-Au-Ag linkage by Olex2. Colors: golden and green, Au/Ag; grey, C; brown, Q peaks. All hydrogen atoms are omitted for clarity.

3. Comment: The percentages of Au/Ag in certain sites seem to be very specific (close to, but not exactly equal to, $1/3$ and $2/3$). (Supp Table 1) Could the authors explain this? Are there any patterns to the Au-Ag positions?

Response: Yes. The percentages of Au/Ag in certain sites seem to be very specific. For example, the occupancies ratios of Au/Ag in the three sites of the fusion M_3 ($\text{M}=\text{Au}$ or Ag) plane are 64.4/35.6, 64.4/35.6 and 37.6/62.4 respectively, which are close to $1/3$

and 2/3. The Au/Ag ratio in the synthesis and the structure geometry of the nanocluster should be responsible for the occupancies of different sites.

For alkynyl-protected Au&Ag alloy nanoclusters, the Au atoms are likely to exist in the surface of metal core due to the nature of coordination chemistry of alkynyl ligands. As shown in Figure R5, the surface alkynyl ligands can bond to metal atoms by σ - and π -type bonding. When the alkynyl ligands adopt the σ -type bonding, the coordinating metal atom must be Au. But for π -type bonding, the coordinating metal atom can be Au or Ag.

Figure R5. The detailed bonding structure of M-alkynyl (M=Au or Ag). Colors: golden, Au; green, Au/Ag; grey, C. All hydrogen atoms are omitted for clarity.

4. Comment: It would be helpful if the atom descriptions given in Supplementary Tables 1 and 2 were accompanied by a figure that used arrows (or similar) to match the descriptions and the positions. This would be especially helpful because the terms used here are not the same as those on p. 4 of the text. I think I have been able to figure out some or all of the positions, but I am not confident in my assignments in relation to the text.

Response: We thank the reviewer for the helpful suggestion. The following description has been included into the main text: “Specifically, $(\text{AuAg})_{31}$ can be described as bi-centered-icosahedra sharing a fusion M_3 face ($13 \times 2 - 3 = 23$), capped with a hexagonal ring of exterior shell Au_6 in the ‘equatorial’ plane of the shared fusion M_3 face and two apical Ag atoms (Figure 2). The metal site Au/Ag disorders, as obtained by least-squares refinement of the occupancies, are indicated in Supplementary Figs. 1 and Tables 1, 2. Finally, the exterior metal shell comprises an Ag-Au-Ag unit (Fig. 3 and Supplementary Figs. 4b, 4c) with one Ag atom bridging one of the nonbonding edges of the ‘equatorial’ hexagonal ring of exterior shell Au_6 (Fig. 3).” Also, in Supplementary

Figure 1, Tables 1 and 2, we have changed “exterior Ag atom” to “apical Ag atom” to make it consistent with the description in other places.

5. Comment: Supplementary Figure 4 has parts listed in the figure, but the caption does not have corresponding letters. The caption seems to refer to two structures, whereas three parts of the figure are given.

Response: Thanks for pointing out the error. The caption is now corrected in the revised manuscript by including the information of the third structure.

6. Comment: Is the monomer to polymer transformation reversible (either by using a different solvent or by heating)?

Response: The transformation chemistry from the monomer to polymer is very interesting. But what a pity is the transformation process is not reversible. We tried different methods to realize the transformation process from the polymer to monomer, but failed to achieve it. The crystals of the polymeric nanocluster cannot be dissolved in any solvent.

7. Comment: The coordinates of relaxed model clusters 1-4 and the polymer should be provided in the SI.

Response: The coordinates are now provided as .xyz files in the SI.

8. Comment: Would the authors have information about the occupied/virtual or valence/conduction band character in this system?

Response: See our response to Referee 3, comments 6 and 9. We have looked at the density of states of a few highest occupied states / unoccupied states (forming the upper edge of the valence band / lower edge of conduction band and the small DOS peaks below/above the bandgap in Fig. 7). The visualizations of the summed DOS of these states are shown in the new Supplementary Fig. 17. We see from the figure, that (i) appreciable DOS overlap between the neighbouring clusters in the polymer exists **only** in the direction of the polymer axis, and (ii) the overlap seems to be slightly larger in the case of the upper edge of the valence band (panel (a)).

9. Comment: It is good that the authors provided the average electrical conductivity over 6 single crystals. It would be best if they present the average and the standard

deviation for the conductivity along the c and a directions (a table in the Supporting Information would be fine) (or perhaps include all six measurements, plus the average and standard deviation) so that readers can know about the spread of the values measured.

Response: Thanks for your suggestion. A table of electrical conductivity, the average and the standard deviation for the conductivity along the c- and a- axis of 6 single crystals has been included as Table S4 in the revised manuscript. The following introduction has been added to the main text: “The averaged electrical conductivity along the c-crystallographic axiswhich is 1800 times of the electrical conductivity along the a-crystallographic axis (values averaged over 6 single crystals). All six measurements, plus the average and standard deviation are shown in Table S4.”

10. Comment: A reference should be included for “AuSMe₂Cl was prepared according to literature methods”. p. 12

Response: We have cited this reference in our revised manuscript as Ref. 59.

REVIEWERS' COMMENTS:

Reviewer #1 (Remarks to the Author):

The authors have addressed all my concerns, and I am very enthusiastic to recommend the acceptance of this paper.

Reviewer #2 (Remarks to the Author):

The authors have answered all questions raised by reviewers and addressed the major issues with the original manuscript. I recommend that the manuscript is accepted.

Reviewer #3 provided comments to the editor only, in which they confirmed they were satisfied with the revisions and recommended publication.

Reviewer #4 (Remarks to the Author):

The authors have generally addressed the comments of the reviewers. One thing that should be changed: The picture of the cluster interactions given by Figure R4 and the surrounding discussion seems potentially quite different than the picture implied by Figure 2. I am not sure what all of the implications could be, but I think that it is very important that this be present in the literature. The authors should include some of their discussion regarding disorder and Figure R4 in the supporting information.

p. 9 bottom: The authors should clarify that the “figure” that shows directionality is the supplementary figure, not the one in the text. (A reader could currently think it is the other figure that was discussed in the same paragraph.)¹²

Reply to Referee # 4

We really appreciate your considerate comments, which are very helpful for us to improve the manuscript. According to your suggestions, we have carefully revised our manuscript as follows:

1. Comment: The authors have generally addressed the comments of the reviewers. One thing that should be changed: The picture of the cluster interactions given by Figure R4 and the surrounding discussion seems potentially quite different than the picture implied by Figure 2. I am not sure what all of the implications could be, but I think that it is very important that this be present in the literature. The authors should include some of their discussion regarding disorder and Figure R4 in the supporting information.

Response: To make it easy for readers to understand the structure of the polymeric (AuAg)_{34n} nanocluster, we have included Figure R4 as Supplementary Figure 6 in the revised manuscript. Structures in Figure 2 and Supplementary Figure 6 are viewed from different directions so that they seem to be different. Actually, Supplementary Figure 6 provides more detailed disordering information for Figure 3. Discussion on the disordered issue has also been included in the caption of Supplementary Figure 6.

2. Comment: p. 9 bottom: The authors should clarify that the “figure” that shows directionality is the supplementary figure, not the one in the text. (A reader could currently think it is the other figure that was discussed in the same paragraph.) .

Response: In the revised manuscript, we have accordingly modified the sentence as “This data is visualized in Supplementary Fig. 11 showing that appreciable density overlap between the neighboring clusters in the polymer exists only in the direction of the polymer axis.”